# The Long Journey from Animal Electricity to the Discovery of Ion Channels and the Modelling of the Human Brain

**DOI:** 10.3390/biom14060684

**Published:** 2024-06-12

**Authors:** Luigi Catacuzzeno, Antonio Michelucci, Fabio Franciolini

**Affiliations:** Dipartimento di Chimica, Biologia e Biotecnologie, Universita’ di Perugia, 06123 Perugia, Italy; antonio.michelucci@unipg.it

**Keywords:** membrane electricity, excitability, action potential, ion channels

## Abstract

This retrospective begins with Galvani’s experiments on frogs at the end of the 18th century and his discovery of ‘animal electricity’. It goes on to illustrate the numerous contributions to the field of physical chemistry in the second half of the 19th century (Nernst’s equilibrium potential, based on the work of Wilhelm Ostwald, Max Planck’s ion electrodiffusion, Einstein’s studies of Brownian motion) which led Bernstein to propose his membrane theory in the early 1900s as an explanation of Galvani’s findings and cell excitability. These processes were fully elucidated by Hodgkin and Huxley in 1952 who detailed the ionic basis of resting and action potentials, but without addressing the question of where these ions passed. The emerging question of the existence of ion channels, widely debated over the next two decades, was finally accepted and, a decade later, many of them began to be cloned. This led to the possibility of modelling the activity of individual neurons in the brain and then that of simple circuits. Taking advantage of the remarkable advances in computer science in the new millennium, together with a much deeper understanding of brain architecture, more ambitious scientific goals were dreamed of to understand the brain and how it works. The retrospective concludes by reviewing the main efforts in this direction, namely the construction of a digital brain, an in silico copy of the brain that would run on supercomputers and behave just like a real brain.

## 1. Galvani and Animal Electricity

This retrospective begins with when scientists discovered that messages in animals were sent from one place to another through nerves and involved electrical processes, in other words, when they discovered ‘animal electricity’. The symbolic figure associated with this time and these studies is Luigi Galvani, Professor of Anatomy at the Academy of Sciences in Bologna, who, in the last three decades of the 18th century, conducted electrophysiological experiments demonstrating that organisms have an intrinsic electricity that they use to send messages throughout the body.

### 1.1. Galvani’s Frog Neuro-Muscular Preparation and the Animal Electricity

Galvani developed the frog neuro-muscular preparation, which soon became a classic experimental model for these studies, consisting of the frog leg muscle (sartorius, most commonly) with the commanding nerve (i.e., sciatic) still attached. The whole thing was detached from the dead frog. Early in the 1780s, Galvani showed that stimulation of the nerve with the Leyden jar, which would deliver an electric shock to the preparation, triggered the contraction of the attached muscle (Figure 1).

Galvani made a careful electrophysiological study of the electrical excitation of the nerve–muscle preparation, varying the conditions and the stimulation protocol, and described all the major features of the observations he made for nearly a decade before finally publishing his results and interpretation of the phenomenon in his famous book, *De Viribus Electricitatis in Motu Musculari* (by the Academy of Sciences in Bologna, 1791) [1]. In his view, the electrical signal was transmitted along the nerve through the internal core that was separated from the extracellular space by an insulating sheath. Notably, this interpretation already contained the notion of a nonconductive cell membrane.

In elaborating his ideas, Galvani was likely influenced by the classic views of Claudius Galen, the Greek–Roman physician of the late second century A.D. who envisioned a ‘pneuma’, a light fluid filling the brain ventricles and the nerve cavities, that would transfer the signals from sense organs to the brain and from the brain to the muscles, and even more so by the French philosopher and mathematician René Descartes’ re-elaboration of the “light wind or … flame”, that he named “animal spirit”, which would run along the nerves to bring messages, like water in a conduit to the fountain [2]. Views that persisted for much of the 18th century thanks to the Dutch physician Herman Boerhaave and several other scholars who continued to propagate the idea of this invisible, weightless fluid secreted by the brain and flowing to the muscles through the nerves [3].

### 1.2. Galvani’s Views Were Not Fully Accepted by the Scientific Community

Galvani’s conclusions were, however, opposed by many scientists from his time on the grounds of him being the one giving the nerve an electric shock, so he could not indeed call it ‘intrinsic’ animal electricity, since electricity was given to the frog from outside [4]. Galvani repeated the experiments, this time by stimulating the nerve with an arc made by two different metals and obtained the same results he had obtained by delivering an electric shock with the Leyden jar.

The new experiments seemed more convincing, but, unfortunately, they came at the time when the physicist Alessandro Volta, who was studying the battery (the pile) at the University of Pavia, demonstrated that electricity was also generated by electrochemical processes occurring in the contact zone of two different metals. Thus, the intrinsic electrical activity that Galvani contended it led to the frog’s muscular contractions was argued against by Volta due to the metals used and the electricity they produced, sustaining the conclusion that it was, therefore, not intrinsic animal electricity. Volta’s arguments appealed to many scientists, who, as a result, began to show skepticism about Galvani’s interpretation [4].

Convinced of his ideas, although now old, Galvani tried to imagine new experiments without using any type of metal to stimulate muscle contraction. He achieved this by immersing the frog leg and the connected nerve into two separate beakers filled with solution. Each time he connected the solutions of the two beakers with moistened paper, the leg contracted (Figure 2A). The same thing happened when the cut surface of the sciatic nerve touched the muscle or when the cut end of the right sciatic nerve touched the intact surface of the left sciatic nerve and vice versa (Figure 2B,C). However, although these new experiments provided convincing evidence of the intrinsic animal electricity, the general excitement over Volta’s battery prevented most scientists from recognizing Galvani’s insights into animal electricity for many years.

### 1.3. Carlo Matteucci Provides the Final Proof of Intrinsic Animal Electricity

It would be the Italian physicist Carlo Matteucci, with his experiments in 1842, after the prejudices that electricity was only produced by Volta’s battery had faded, who would conclusively demonstrate the intrinsic electricity of animals. Using his improved galvanometer, and putting one electrode on the intact muscle and the other on the cut end, he could record a small “current directed from the inside of the wound to the outside surface of the muscle” [7]. This current will be named “injury currents” in 1870 by Ludimar Hermann [8], a student in du Bois-Reymond’s laboratory (see later).

However, as the current from one single cut muscle was extremely small, inspired by the multiple disc structure of the fish electric organ and Nobili’s suggestions, Matteucci thought that bigger currents could be detected by cutting several frog muscles and assembling them serially with the intact part of one muscle partially inserted into the cut end of its neighbor. The current recorded from this composite preparation was found to increase in a stepwise fashion, in parallel with the addition of each new cut muscle to the pile, while the number of metal–tissue contacts made by the electrodes (two) remained the same [9].

These studies were continued by du Bois-Reymond using more sensitive galvanometers that also allowed him to investigate the electrical aspects of nerve fibers [10]. His studies established that cut ends of muscles and nerves were negative compared to intact areas, and, in general, recognized the correctness of Galvani’s animal electricity. In retrospect, we can also say that Galvani’s legacy is much broader than the observations he made and the conclusions he reached, because it concerns his new approach to tackling biological problems with the laws of physics and chemistry: a watershed in the study of the natural sciences (in the German meaning of ‘Wissenschaft’). Readers interested in the topic of animal electricity are referred to the following fine reviews on the subject: [3,4,11,12,13,14].

## 2. Bernstein and the Membrane Hypothesis

The next challenge was to identify the structures and the mechanisms underlying animal electricity and the injury current. A first attempt to address these issues arrived in 1902 with the German physiologist Julius Bernstein and his membrane hypothesis [15]. In formulating his theory, he greatly benefitted from major achievements in the field of physical chemistry and in the structure of biological membranes gained in those years.

Crucial help came from Walther Nernst’s equilibrium potential, based on the work of German chemist–physicist Wilhelm Ostwald who had studied semi-permeable artificial membranes and the electrical potentials formed across them as a result of the membrane’s selective permeability to ions [16]. Nernst’s thermodynamic analysis of Ostwald’s diffusion potentials led to Nernst’s electrochemical equation [17], which proved to be crucial for the maturation of Bernstein’s membrane hypothesis. Also important were Max Planck’s results on electrodiffusion according to which ion currents derive from the combined contribution of diffusional and electrophoretic motion of ions [18]. In this context, we cannot fail to mention Einstein’s studies of Brownian motion as the random walk of molecules that provided a physical interpretation of the diffusion coefficient in the Nernst–Planck equation [19].

Both Ostwald’s and Nernst’s concepts apply to ion fluxes across membranes, of which, unfortunately, only a vague idea existed about their nature and structure in Bernstein’s time, not to mention that many were intimately skeptical of the very notion of membrane-enveloped cells. However, Hans Meyer, in 1899, and Charles Overton, in 1901, while studying the efficacy of anesthetics, found a strong correlation between the lipid solubility of the anesthetics tested and their potency, a discovery which led them to conclude that lipids of some kind must be the basic components of the cell membrane [20,21].

### 2.1. The Membrane Hypothesis of Julius Bernstein

Based on the now established knowledge that the main ions inside and outside nerves and muscles are distributed very unevenly, thus forming strong gradients, and on the physico-chemical work by Ostwald and Nernst, Bernstein proposed that the injury potential, i.e., the electric potential difference across the cell membrane causing the injury current, was the combined result of the K^+^ gradient across the membrane and its selective permeability to K^+^ ions.

Bernstein tested this notion by checking if the injury current increased linearly with temperature, as predicted by the Nernst equation. To this end, he immersed a cut frog muscle in oil, with one recording electrode placed in contact with the muscle cut end and another recording electrode in contact with the intact external surface, and put a thermometer in contact with the muscle. He then changed the temperature of the bath and found that the injury current increased linearly with it, as predicted by the Nernst equation, giving strong support to the notion that cells were surrounded by a membrane essentially impermeable to all ions except K^+^. This occurrence, combined with the observation that K^+^ ions are about 30 times more concentrated inside the cell than outside, would create a potential difference across the membrane, with the inside negative, from −70 to 90 mV depending on the K^+^ gradient, which only became apparent in the event of injury. These were the main points of Bernstein’s membrane hypothesis [22].

Bernstein’s hypothesis, essentially based on K^+^-selective membranes under resting conditions, not only explained the resting potential, but could also explain the action potential (see below). One only had to assume that, during excitation, the membrane would transiently lose its selective permeability to K^+^ and become permeable to all ions. This would result in a rapid fall of the membrane potential to zero, in accordance with the transient decrease in the injury current, a reflection of the membrane potential which he had observed more than 30 years earlier when the muscle was electrically stimulated to contract.

### 2.2. Bernstein’s Recording of the ‘Negative Variation’ or Action Potential

If Bernstein’s most important accomplishment remains the proposition of the membrane hypothesis, his recording, for the first time, of the ‘negative variation’, i.e., the action potential ante litteram, should not be forgotten. To appreciate the importance of Bernstein’s observation of the negative variation, a brief historical background summary may be useful.

Ten years after Matteucci had observed a current flowing from the cut surface of a muscle preparation to the intact section (i.e., from the inside to the outside), du Bois-Reymond observed a transient decrease in the injury current when the muscle was electrically stimulated to contract and called it a negative variation [10]. Because of the poor time resolution of the instruments available at the time, it was impossible to establish whether the negative variation mirrored (i.e., had the same time course of, for instance) the excitatory process.

To address this question, Bernstein constructed a high-speed differential rheotome or ‘current slicer’ capable of following the changes of the injured potential following nerve excitation at a high resolution (Figure 3, right). He found that the potential developed rapidly and consisted of an initial depolarization followed by a repolarization with a slight slower tail (Figure 3, left). This can be taken as the first recording of a nerve action potential. Of note, he also observed an ‘overshoot’, yet he did not emphasize this observation because he suspected that it could be due to junction potentials or other artefacts. So much so that, when he formulated the excitation hypothesis in 1912 [22], he proposed that, during cell excitation, the membrane became indiscriminately permeable to all ions and the membrane potential dropped to zero. He also determined the conduction velocity of the impulse propagation at ~30 m/s, in perfect agreement with Hermann von Helmotz’s previous estimates of 25–40 m/s [23].

At about the same time that Bernstein was elaborating his membrane theory, Ramon y Cajal provided the conclusive demonstration that neurons are single cells, distinct and separate from their neighbors, and represent the structural and functional units of the nervous system [25]. These observations put an end to a long-standing dispute that had arisen in the late 19th century over the organization of the nervous system, namely the ‘reticular theory’ proposed by Camillo Golgi, according to which neurons are connected in a single superordinate syncytium (cf. [26]). Cajal’s proposal that nerve cells were not continuous in the brain was not fully accepted until several decades later, when electron microscopy showed that neurons are separated by the presence of synaptic clefts [27,28]. In a way, the neuron doctrine appeared to retrace the events that brought Robert Hooke and Antoni van Leeuwenhoek, in the second half of the 17th century, to the identification of the cell as the unitary, basic element of which all living organisms are made, in other words, the cell theory.

### 2.3. The Evolution of Membrane Models

Meanwhile, new concepts and a more precise definition of cell membrane architecture were rapidly developed. The initial view of the membrane as mainly lipoid in structure, proposed by Meyer and Overton at the turn of the century, was much better defined in 1925 by the Dutch scientists Evert Gorter and François Grendel. By comparing the total surface area of red blood cells from different animal species and the area of the lipid monolayer that the considered red blood cells generated (assessed with the Langmuir’s method), they found the monolayer to be twice as large as the cumulative surface area of the red blood cells used [29]. Based on this observation, they concluded that the cell membrane was organized as a lipid bilayer, with their polar heads facing either the outside or the inside of the cell, and their non-polar tails facing each other at the center of the matrix. However, the model did not answer two main questions. First, how could lipid-insoluble molecules enter the cell if the membrane consisted exclusively of lipids? Secondly, how to explain the much lower surface tension of the membrane compared to that of pure oil?

To overcome these shortcomings, in 1935, Hugh Davson and James Danielli suggested that the lipid bilayer was covered on both sides by a protein layer, something which would explain the low surface tension of the membrane. The sandwich model, as it came to be known, obtained strong support from the first electron microscope pictures of the cell membrane that started to appear in the mid-1950s, showing the membrane as being made up of a light central structure, the hydrophobic lipid tails, covered by two dark regions, misinterpreted as being due to the protein layers, in addition to lipid heads [30].

## 3. The Prelude to the Great Breakthrough (1930–1952)

After the Bernstein years, the question of cell excitability laid dormant for a while until it was taken up again in the mid-1920s by Kennet Cole, who began to study the properties of cell membranes by measuring, together with his postdoctoral researcher Howard Curtis, their impedance in the giant alga *Nitella* and in invertebrate eggs using the Wheatstone bridge. Their studies were also meant to test Bernstein’s hypothesis of membrane excitation. Inspired by the axon cable theory proposed years earlier [31], they modelled the cell membrane as a resistor–capacitor circuit. From these measurements, they concluded that the membrane consisted of a very thin structure with a high resistance to ions and high capacitance. However, these data were insufficient to test the hypothesis that cell excitability was the result of voltage-dependent permeability changes. To do this, it was necessary to make intracellular measurements, which proved to be a prohibitive task at the time because the electrodes, still pulled by hand, were too large to be inserted into conventional cells without irreparably damaging them.

The solution came with the introduction, in the summer of 1936, of the giant squid axon, thanks to the suggestion by the young British zoologist John Young who was studying cephalopods at the Zoological Station of Naples. These axons are cylindrical structures up to 1 mm in diameter into which it was fairly easy to insert hand-made glass electrodes. Rarely has the introduction of a new cell model had such a great impact on the progress of science. For some forty years, it was the workhorse of membrane biophysicists. Then, as quickly as it appeared, the giant squid axon disappeared, practically vanishing from the laboratories within a few years, supplanted by new cellular models that the giga-seal patch clamp technique made possible to electrically access and record from.

### 3.1. Membrane Impedance Decreases 40-Fold during Excitation

Using the giant squid axon, Cole and Curtis demonstrated the breakdown of membrane resistance during excitation by simultaneously measuring the membrane impedance with electrodes connected to a Wheatstone bridge and the change in membrane potential with extracellular electrodes. When the excitation invested the axon, the signal on the scope, typically a line at resting condition, transiently broadened into a large band due to bridge unbalance (Figure 4; [32]). Analysis of the band showed that the membrane resistance during excitation decreased about 40-fold compared to rest. Equally important, the amplitude of the band (i.e., the reciprocal of membrane impedance) varied in parallel with the time course of the excitation (white trace), thus providing evidence of the strict relationship between the two processes. Another important observation was that the membrane impedance began decreasing only after the membrane potential had been significantly depolarized from the resting potential. Unfortunately, these experiments could not provide a quantitative measure of the effective membrane potential of the axon, because the conventional external electrodes used could only record a fraction of the actual transmembrane potential. Nonetheless, Cole and Curtis’s results were fully consistent with Bernstein’s hypothesis of cell excitability.

### 3.2. The Action Potential Recording Shows an Unexpected Feature: The Overshoot

When cell excitability seemed essentially explained, in the summer of 1939, Alan Hodgkin and Andrew Huxley, using the giant squid axon inside which they could easily insert handmade electrodes of about 100 µm in diameter, made the first quantitative recordings of the action potential. The action potential recorded by Hodgkin and Huxley recalled very closely the ‘negative variation’ (the negative ‘Schwankung’) recorded 70 years earlier by Julius Bernstein. Most importantly, however, the use of intracellular recording allowed Hodgkin and Huxley to obtain very reliable quantitative measurements that, quite unexpectedly and without any doubt, showed that the potential drop did not just go to zero, as predicted by Bernstein’s membrane hypothesis, but went far beyond this, reaching positive values of up to 40–50 mV (the ‘overshoot’) (Figure 5, left). A new mechanism for cell excitation and the action potential was needed. But the Second World War, which broke out on 1 September 1939, interrupted all further investigations. Hodgkin and Huxley only had time to send a short letter to *Nature* which contained only the figure for the action potential and a minimal description of the methods and results; there was no discussion [33].

At the end of the war, Hodgkin and Bernard Katz (as Huxley was temporarily engaged in personal matters) returned to Plymouth in search of an explanation for the overshoot. Their reasoning was simple: given the high concentration of external Na^+^ and the overshoot reaching values close to the Na^+^ equilibrium potential, they thought that the membrane became transiently and selectively permeable to Na^+^ ions during excitation, and not indiscriminately to all ions. In this way, Na^+^ would cross the membrane along its concentration gradient and depolarize it to its equilibrium potential, about +50 mV. Hodgkin and Katz tested this hypothesis by replacing external Na^+^, or part of it, with large ions of choline, presumably unable to cross the membrane, and observed that the action potential rose much more slowly and the overshoot became progressively smaller the more Na^+^ was replaced [34] (Figure 5, right). Consistent results were obtained with radiolabelled Na^+^, showing a significant transmembrane flux of the ion during the action potential [35]. To complete the picture, an experimental visualization of the Na^+^ current across the membrane was now needed.

## 4. Understanding and Modelling the Action Potential

To test the Na^+^ hypothesis, i.e., the voltage-dependent change in Na^+^ permeability and the resulting Na^+^ current, it was necessary to record the current, as a readout of membrane permeability, at varying constant voltages. This is what George Marmont, a postdoctoral researcher in Kenneth Cole’s laboratory, set out to do immediately after the war. He began to develop a feedback system (the voltage clamp) that allowed the membrane potential of the giant squid axon to be controlled and maintained at a constant level to record membrane currents at chosen membrane potentials [36,37]. This new method made it possible to record the membrane currents under voltage clamp and provide the first description of the electrical properties of the membrane and the ionic conductances underlying the action potential.

### 4.1. Hodgkin and Huxley’s Recordings of Na^+^ and K^+^ Currents

These first voltage clamp recordings were not perfect, as Hodgkin noticed on a visit to Cole’s laboratory in the spring of 1948. In particular, his attention was attracted by a systematic delay at the onset of the current, at the beginning of depolarization, which appeared to be at odds with earlier simulations Huxley had made of the action potential. Back in Cambridge, Hodgkin and Huxley guessed that the delayed onset of the current observed in Cole and Marmont’s recordings was due to slow electronics, possibly caused by too high a series resistance of their internal electrode. For this reason, Hodgkin and Huxley used a double electrode which eliminated the problem and worked better for series resistance compensation. This proved to be a great improvement over the device used by Cole and Marmont. In short, we can credit Marmont and Cole for coming up with the idea of the voltage clamp and developing the first system [36,37], but it was Hodgkin and Huxley who perfected it and exploited its full power.

In the summer of 1949, Hodgkin and Huxley, building on the experience of the previous experimental season, performed all the voltage clamp experiments and recorded all the currents that they used to describe and interpret the ionic basis of the action potential and its propagation that appeared in their 1952 papers [38,39,40,41,42]. A representative example of a current recorded by Hodgkin and Huxley upon a depolarization from near resting potential to 0 mV under a voltage clamp is shown in Figure 6. The current has a complex time course, with an initial inward phase followed by an outward component. From ion substitution experiments and by using different voltage clamp protocols, they could establish that the initial inward and subsequent outward current components were due to Na^+^ entry into the axon and K^+^ exit, respectively, and concluded that the two currents were the expression of independent permeability mechanisms, with membrane conductances for Na^+^ and K^+^ changing as a function of time and membrane potential.

Noting that changes in Na^+^ and K^+^ currents (and conductances) depended on changes in membrane potential, Hodgkin and Huxley’s first conclusion was that there must be membrane-bound charged particles or dipole structures that moved across it as the potential changed, thus controlling the Na^+^ and K^+^ fluxes. This concept, inescapably dictated by the experimental results, was pivotal in Hodgkin and Huxley’s modelling of the membrane currents and the action potential of the giant squid axon, as we will see now.

### 4.2. Hodgkin and Huxley’s Model of the Action Potential

Hodgkin and Huxley tried to model mathematically the ionic processes underlying the action potential and provide a quantitative model of cell excitability of nerve cells. One observation they soon made was that both the Na^+^ and the K^+^ current’s onset in response to a depolarization did not follow first-order kinetics, but showed a sigmoid-shaped increase after an initial delay, whereas, upon repolarization, they fell exponentially toward zero, and much faster than they rose. This feature was systematized by working on the charged particle concept mentioned earlier. That is, they assumed that more than one of these membrane-bound charged particles was associated with a not better-defined membrane permeation spot or patch, and that they all had to move through the membrane under the influence of the electric field from a non-permissive to a permissive position to allow Na^+^ and K^+^ ions to pass. On the contrary, it was sufficient that one of them moved back to the non-permissive position upon repolarization to make the permeation spot impermeant to ions.

With this idea in mind, Hodgkin and Huxley tried to write separate equations for the Na^+^ and K^+^ conductances, *g_Na_* and *g_K_*, respectively, and their voltage and time dependences [40]. They expressed them as having an upper limiting value, *g_Na,max_* and *g_K,max_*, and their actual value at each given voltage and time being determined by coefficients, varying between 0 and 1, represented the fraction of the membrane-bound charged particles in the permissive position. They denoted these charged particles as *m* and *n* for Na^+^ and K^+^ current activation, respectively. In analogy to the activation kinetics of the two conductances, they did the same for the inactivation process of the Na^+^ conductance, assuming that one charged particle, *h*, was governing the process. Hodgkin and Huxley, then, derived dedicated equations, which were matched to their voltage clamp data to determine the voltage dependence of *n*, *m*, and *h*.

Using a classic (chemical) kinetic approach to describe the voltage dependence of the Na^+^ and K^+^ conductances, and adjusting the number of gates, *m* and *n*, controlling the Na^+^ and K^+^ conductances (as well as their maximal values), Hodgkin and Huxley were able to describe very well the Na^+^ and K^+^ currents, assigning a value of 3 to *m*, the charged particles controlling the Na^+^ conductance, and a value of 4 to *n*, controlling the K^+^ conductance. Only one charged particle *h* was used for the Na^+^ inactivation process.

More specifically, the Na^+^ and K^+^ currents could be nicely described by the following equations:$$I_{Na}=g_{Na}\;\left(V-E_{Na}\right),\;\mathrm{with}\;g_{Na}=g_{Na,max}\;m^{3}\;h$$
$$I_{K}=g_{K}\;\left(V-E_{K}\right),\;\mathrm{with}\;g_{K}=g_{K,max}n^{4}$$
where *V* is the membrane potential difference and *E_Na_* and *E_K_* are the Na^+^ and K^+^ Nernst equilibrium potentials, respectively. The fraction of charged activation and inactivation particles, *n*, *m*, and *h*, were described according to first-order chemical reactions using the following differential equations:$$\frac{dn}{dt}=\alpha_{n}\;\left(1-n\right)-\beta_{n}\;n$$
$$\;\frac{dm}{dt}=\alpha_{m}\;\left(1-m\right)-\beta_{m}\;m$$
$$\frac{dh}{dt}=\alpha_{h}\;\left(1-h\right)-\beta_{h}\;h$$
where *α* and *β* represent the transition rate constant for the permissive and non-permissive positions of the charged particle, whose voltage dependence could be experimentally determined.

Using these equations, and considering the other currents that, in addition to the described Na^+^ and K^+^ currents, appeared upon voltage changes, namely the leak current $g_{L}\left(V-E_{L}\right)$, where *E_L_* is the reversal potential of the leak current, and the capacitive current generated by the application of the voltage step $C_{m}\;\frac{dV}{dt}$, where *C_m_* is the membrane electrical capacitance, Hodgkin and Huxley modelled the membrane potential dynamics using the following differential equation:$$\frac{dV}{dt}=\frac{I_{C}-I_{tot}}{C_{m}}$$
where *I_C_* represents the command current imposed by the experimenter.

Notably, the model could accurately predict the time course of the action potential (Figure 7). It also reproduced, and implicitly explained, the action potential propagation, threshold, and refractoriness, that is, it explained the basic mechanism of cell excitability.

### 4.3. The Fallout of Hodgkin and Huxley’s Model

Hodgkin and Huxley’s model not only correctly reproduced and explained the main features of the squid axon’s excitability, but also, by conclusively showing the passage of ions through the membrane during excitation, it made more urgent the need to address the question of where and how ions crossed the membrane. This task was seriously undertaken by Bertil Hille and Clay Armstrong, who were able to define, over the full spread of the 1960s, that the Na^+^ and K^+^ pathways were distinct molecular entities with selective pores of different sizes. After defining the main properties of these ionic pores, their fluxes and their basic biophysics and pharmacology, Hille and Armstrong began to call them ‘channels’, during a time when the scientific community was rather skeptical about the concept. The restricted passage along the Na^+^ and K^+^ channels was estimated to have a size of 3–5 Å and assumed to be lined by oxygen dipoles that would establish hydrogen bonds with the permeating ions [43,44,45,46]. In support of the assertion that they were separated, for the two ions, there were the observations that tetraethylammonium (TEA) blocked the K^+^ current without affecting the Na^+^ current [47,48], whereas tetrodotoxin (TTX), a potent neurotoxin commonly found in marine fish, selectively blocked the Na^+^ current without any effect on the K^+^ current [49]. This view of separate pathways through the membrane for Na^+^ and K^+^ was not easy to accept, considering the scant supporting evidence and the strong skepticism on the part of most scientists.

It took Seymour Singer and Garth Nicolson’s fluid mosaic membrane model showing integral membrane proteins in the membrane [50], Erwin Neher and Bert Sakmann’s single-channel recordings showing elementary Na^+^ and K^+^ currents [51], the effect of proteolytic enzymes on the excitability of the giant squid axon membrane and inactivation of the Na^+^ channel [52,53,54], and the extraction and purification from electric eel electrocytes of a TTX-binding protein [55], which was later found to conduct Na^+^ when inserted into liposomes [56], for ion channels to be fully accepted as protein molecular entities for the selective and gated passage of ions across the membrane.

### 4.4. Hodgkin and Huxley’s Formalism in Modelling Neuron Excitability

Hodgkin and Huxley’s formalism clearly showed its capabilities to faithfully reproduce the time course of the action potential and several other properties such as onset timing, firing threshold, refractoriness, and conduction velocity, and, as such, represented the first successful modelling of the cell excitability. Indeed, rather than modelling the excitability of an entire neuron, they did so only for a short cylindrical segment of the giant axon of the squid, severed from the entire neuron, which is instead made also of dendrites, cell soma, and the initial segment, only to mention the main parts. Moreover, the organism from which they derived the specimen on which to work, i.e., the squid, is an evolutionarily ancient structure carrying, in its genome, only a limited number of genes for ion channels compared to the more than 50 genes estimated to be present in mammals for the pore-forming subunits of voltage-gated channels. Hodgkin and Huxley were, in fact, able to model the action potential of a squid axon segment remarkably well by including only two ion channels, more precisely two currents, i.e., the fast-activating and inactivating Na^+^ current and the delayed rectifying K^+^ current. Modelling an entire neuron of fully evolved organisms such as mammals with a much larger number of channel types, which evolution has generated to allow a fine control of ion flows and action potential shapes, and doing so for an entire neuron composed of many different parts, each with a different repertoire of channels, is a much more challenging undertaking.

Although complexity increases enormously when modelling an entire mammalian neuron containing dozens of voltage-gated channels, differently modulated by multiple cellular pathways and variously distributed in different districts of the neuron, for decades, in order to model neurons, scientists used the voltage- and time-dependent conductance model of Hodgkin and Huxley and their differential equations (cf. [57,58,59]). In this way, they finely modelled many different types of neurons considering their repertoire of ion channels, receptors, and transporters, as well as their localization and density in their various districts. Prominent examples are the CA1 pyramidal neurons of the hippocampus and the cerebellar Purkinje cells and Golgi cells. The use of this knowledge, implemented with appropriate physico-chemical principles and the continuous cross-checking of the model results with experimental data, either generated ad hoc or retrieved from the literature, allowed robust models to be obtained which were capable of faithfully reproducing the experimental data of neuronal excitability [60,61,62,63]. Around the same time, however, neuronal cell firing was also beginning to be reconstructed in silico, using simulation methods, as part of the larger project of constructing a digital copy of a rodent, to start with, and, then, of the human brain.

## 5. Modelling the Human Brain

Understanding the human brain has been the great quest of mankind over the centuries and a major scientific goal of science today. The investigation of the human brain moved in steps, from the level of single neurons to the modelling of functional neural circuits and then of single brain areas, as stepping stones to eventually model (and understand) the whole brain. If, in principle, the study of the brain structure and the understanding of its functioning may seem an attainable task because the basic molecular and cellular mechanisms and the physico-chemical principles underlying its working are essentially the same as those associated with its building blocks (single neurons, neural circuits, and so on), the huge complexity of the brain should not be forgotten, with its almost 100 billion neurons grouped into more than a hundred major types, and a thousand times as many synapses wiring them together, synapses that are themselves representative of many different types.

This enormous complexity initially made the enterprise of modelling the brain seem hopelessly pointless, and, even today, many scientists keep seeing it this way. For this reason, the first steps in the study of the brain were marked by a reductionistic approach, that is, investigating the elementary components of the brain, as neurons and the membrane structures (mainly ion channels) underlying their electrical activity, as we have seen, or simple neural circuits at most. Decades of recording membrane potentials and ionic currents of brain neurons had produced a wealth of biophysical and functional data and important conceptual advances, while anatomo-structural studies had begun to delineate the architecture of increasingly large areas of the brain. Nevertheless, it was also becoming increasingly clear that, due to the brain complexity, this approach—the idea to model and simulate the working brain only through the information and knowledge gathered with the reductionistic approach—appeared unrealistic and not suitable to lead to an understanding of how the brain worked. It is for these reasons that a handful of scientists at the intersection of neuroscience, computer science, and information technology have begun to discuss a new and comprehensive change of perspective to address this issue, consisting of the construction of a digital copy of the brain using the structural and functional information already available and hoping the model would aid the discovery of new ones.

### 5.1. The Strategy of the New Approach

The direction to take was the construction of a digital copy of the brain, a digital brain that would run on supercomputers and behave just like a real brain [64]. Neurophysiologist Henry Markram was the one who envisioned this new course of action and conceptualized the idea of constructing a digital copy of the brain. He believed that it was possible to construct a digital brain without having all the necessary knowledge to model the brain in a conventional way beforehand. On the contrary, the idea came about precisely because of the belief that knowledge would never be gathered. In fact, the expectation was that the digital brain would help identify the still missing data and provide the knowledge that was not yet there, and this would be done while the construction of the digital brain was in progress.

The way this idea was expected to be implemented was to construct an initial digital framework of a small functional structure of the brain into which all information and data known at the time, as a result of laboratory experiments, clinical investigations, or theoretical analysis, would be collected. It was also crucial to provide this structure with the available basic rules and principles of how biological elements and processes function, i.e., the types of neurons present and how they are distributed in specific brain substructures, how they interconnect with each other through synapses, the main ion channels they express, and so on.

At the beginning, the available data and the rules relating them were expected to be limited compared to the many missing elements. However, this was not viewed as a major obstacle but, rather, as the logic of the strategy. Using the initial information, the algorithm would attempt to dislocate the various types of neurons in the brain, equip them with the excitability profile, and link them synaptically in accordance with the instructions and principles given. The output, in the form of firing mode, excitation pathways, and synaptic integration, would be compared with a set of dedicated data left on the side for this purpose (i.e., not used as initial data seed to make the algorithm). If the comparison was satisfactory, more data were added so that the system could grind and continue to build a more accurate model. Otherwise, it would be necessary to go back and see what went wrong with the rules and principles provided or assess if more data were needed.

A great deal of skepticism toward this ambitious—absurd, in the mind of many—undertaking was encountered at the time it was initially proposed, because it was thought that no one would ever be able to correctly reproduce the connectivity of such a large number of neurons and synapses. Neuroscientists were the most skeptical, believing that large-scale simulations made little sense if they were not constrained by experimental data or used to test specific hypotheses. In fact, they never thought of going in that direction. But was this a sufficient motivation for not trying?

### 5.2. Modelling the Whole Brain

At the turn of the century, the idea began to take hold, especially the conviction that, to model the whole brain, it was not necessary for all the information to be known and the wiring to be made, or for all the secrets of the brain to be known beforehand. Henry Markram proposed this idea to the Swiss research authorities and, in 2005, the Blue Brain Project was launched under his direction, in collaboration with IBM (the Big Blue), with the aim to reconstruct, in silico, the brain of a mouse).

When they started, they were only able to model in silico little more than single neurons and small neuron networks. Nonetheless, the first goal they set themselves was to model a complex and highly evolved structure of the mouse brain: a neocortical column, a millimeter-sized cylinder of tissue made up of several tens of thousands of different types of cells, where our sensations, actions, and consciousness reside. This columnar structure can be considered the elementary functional unit for many types of information-processing tasks that evolution has perfected and replicated, with small variations, in virtually all brain regions. They gathered as much information as they could about the layered structure of the column, the types of neurons present, the properties of each single type, and fed them to a supercomputer, while allowing each virtual neuron to connect with the others in all possible ways among those found experimentally.

Three years later, in 2008, Markram and co-workers tested the electrical behavior of the digital neocortex column as if the same structure were to be tested in a mouse cortical slice on an electrophysiological setup (Figure 8). Following the application of a conventional triggering stimulus, spikes began to appear with the typical shape of those recorded electrophysiologically, propagating radially and between the column’s layers in a seemingly endless fashion [65].

Blue Brain scientists later implemented the model by including astrocytes connected with the neurons to reflect more closely the actual brain structure and published the first digital copy of a neocortex column [66]. They also found that brain processes operated on many more dimensions than the three we were familiar with [67], and this could be the reason why, in their view, it is so hard to understand the brain. As a compendium of their studies, they also made the first digital 3D brain cell atlas of the mouse, an interactive and dynamic online resource that provides information on every cell in the mouse brain (cell types, numbers, position in more than 700 areas of the brain, properties) [68].

In 2013, Henry Markram prepared a large project for the European Commission at the time when it was deciding on the allocation of large fundings for two flagship projects. The Commission bet on his dream of building a detailed copy of the human brain on a supercomputer and funded the 10-year-long Human Brain Project (HBP). In several respects, the HBP was an expansion of the work started in 2005 on the mouse with the Blue Brain Project, replicating essentially the same core strategy; no wonder, then, that it was headed by the same Henry Markram who directed the Blue Brain Project.

#### Multiscale Approach

A data-driven model of the complexity of the human brain must also take into account the fact that the processes under investigation occur at scales differing by many orders of magnitude with respect to both length and time. Missing details when studying the higher scale or the big picture when focusing on details may become a real problem. To avoid this, different modelling approaches were planned for use and integrated to reproduce what happens at different scales and then link them together in the so-called multiscale simulation approach [69,70]. The reference here is specifically to the integration of the ‘bottom-up’ approach, which models the submolecular and molecular level of membrane transporters which are then transferred as averaged information to the next higher scale level, the synapses, neurons excitability, and neuronal circuits of increasing complexity, and so on up to macroscopic brain observables and functions, with the ‘top-down’ approach, which starts from the macroscopic observations of the intact brain, such as MRI, EEG, MEG, PET, and from their parameters infers anatomical networks and neuron activity that would explain those observables [71,72,73] (Figure 9).

### 5.3. Accomplishments of the Human Brain Project

Now that the HBP is over, let us take a look at its accomplishments. For one thing, the project introduced a new paradigm in brain research: the integrated approach between computer science and neuroscience, made possible by the recent developments in computer science and big data management and analysis. This has enabled the development of multiscale computational models to simulate neuronal activity and brain function from basic experimental data in genetics, molecular biology, cellular biophysics, and brain imaging. Never before have computer and information technologies entered so heavily the study of the human brain and greatly helped develop unique new research infrastructures, examples of which are given below. One is the high-resolution brain atlas, an extremely detailed digital device reporting information on neurons, their connections, and functional specializations in the brain at a very high resolution. The EBRAINS Research Infrastructure, a high-computing-power resource made available to the brain research community to run a wide variety of virtual experiments, is a further example, greatly accelerating research results, sparing animals from experimental procedures, and reducing research costs. Lastly, the Spiking Neural Network Architecture, or “SpiNNaker”, a neuro-inspired computer machine based on the notion that spiking neural networks can be transformed in hardware to generate neuromorphic computing machines and autonomous robots. In addition to the above, the publication of over three thousand high-impact papers during the course of the project substantially contributed to our understanding of the human brain.

On the other hand, if we evaluate the results achieved by the project against the initial stated goal, which was to build a detailed and realistic computer model of the human brain, our judgement might be different. Although we are educated enough to understand that that goal was something that everyone, even the proponents and the funding committee, knew was impossible to achieve and was proposed only to make a great impact on the general public and European Commission, the distance between the stated goal and the results obtained remains considerable. In fact, we have certainly not witnessed a revolution in the conceptual knowledge of the human brain, as one could rightly have expected from a ten-year project involving more than 500 top scientists and funded with more than 600 million euros.

## 6. Conclusions and Outlook

Some 250 years ago, Luigi Galvani demonstrated that animals have intrinsic electricity. Today, we know that, of all tissues, it is the brain that handles that electricity at its highest level of sophistication in terms of management and results achieved. It is the electricity in the brain that makes us see and hear, that moves our muscles to speak and walk, that commands many other actions that keep us alive. The great complexity of this orchestration has pushed the evolution of the brain to such a level of complexity that recent research has demonstrated that, only to begin to understand its rudimental aspects, enormous efforts are required.

Besides, the electricity of the brain is by no means just that. It is much more. The electricity of the brain is what makes us think, feel emotions, make decisions, learn things, have memories from our past, imagine our future. All these functions—if we can call them that—are certainly and strictly linked to the brain by a relationship that is impenetrable for us to understand at the moment. In other words, we do not know how they are related to the brain architecture, to its biophysical processes, or higher-level observables. These functions are what ancient philosophers and then neuroscientists have come to call ‘mind’. With regard to our understanding of the relationship between the electricity in the brain and the activities of the mind, we are still, in relative terms, in the stone age.

We should then ask ourselves whether we will ever be able to uncover all the mysteries of the human brain. The keen observation that physicist Emerson Pugh made about 100 years ago rings like a death knell: “If the human brain were so simple that we could understand it, we would be so simple that we could not understand it [75]”. By reversing logic, our brain will always be more complicated than it, the brain itself, is capable of understanding. However, it might not be bad news to be confronted with, from time to time, our own limitations.

## Figures and Tables

**Figure 1 biomolecules-14-00684-f001:**
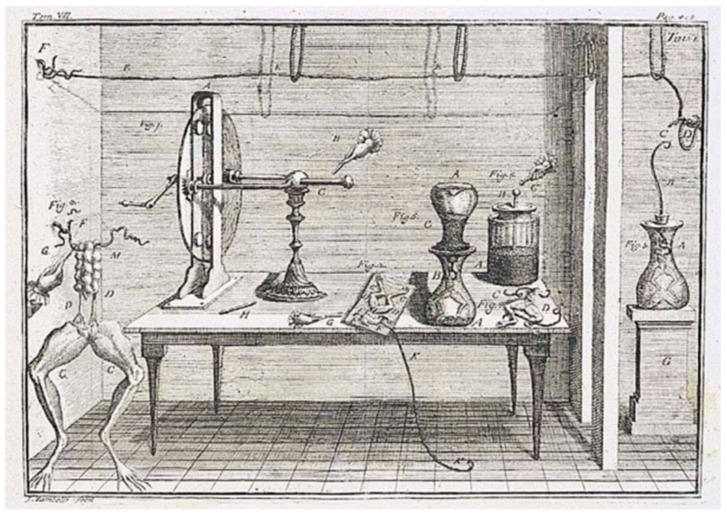
**Classic experiments showing animal electricity and the instruments used at the time.** Galvani noticed that the muscles of a frog leg twitched and contracted when a spark generated by the Layden jar was delivered to the nerve. He claimed that the contraction of the muscle was due to animal electricity propagating through living tissues in the body. (From [1]).

**Figure 2 biomolecules-14-00684-f002:**
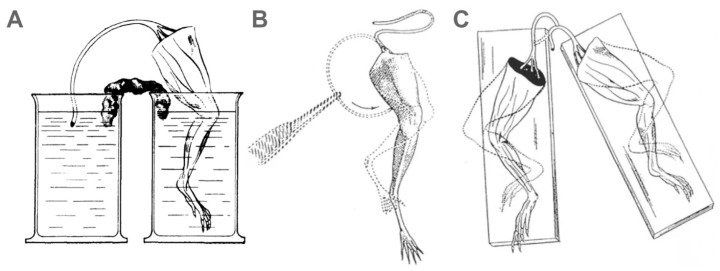
**Galvani’s experiments without the use of metal to stimulate muscle contraction.** (**A**) The frog leg contracts when the cut sciatic nerve is made to touch the muscle through salt solutions and wet paper. (**B**) The frog leg contracts when the cut surface of the sciatic nerve touches the muscle. (**C**) Both frog legs contract when the cut end of the right sciatic nerve touches the intact surface of the left sciatic nerve. (Modified from [5,6]).

**Figure 3 biomolecules-14-00684-f003:**
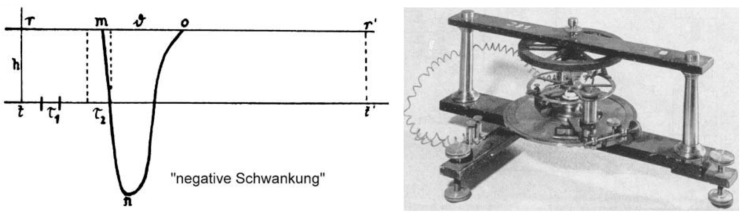
**Bernstein’s first recording of an ‘action potential’ from the nerve.** (**Left**) Reconstruction of the time course of the ‘negative variation’ (negative Schwankung), i.e., the ‘action potential’ ante litteram. t, time axis; t1 and t2, sampling intervals; r m, latency time; m o, duration of ‘negative variation’ (n). (**Right**) The Bernstein rheotome used to record the first ‘action potential’ from the nerve. (From [24]).

**Figure 4 biomolecules-14-00684-f004:**
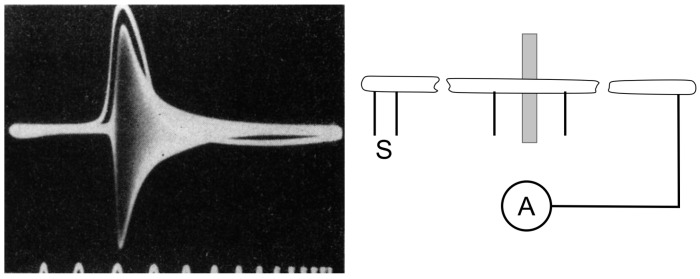
**Impedance change and excitation time course of the squid giant axon membrane recorded using the circuit shown on the right.** (**Left**) Output from the Wheatstone bridge during axon excitation. The bridge was initially balanced with the oscillograph at rest to give a narrow horizontal trace each sweep. When the axon was excited, the bridge went off balance and the oscillograph line broadened into the band shown. Then as the axon excitation subsided, the band narrowed down to the resting line again. The white trace is the recording of a monophasic potential. (**Right**) Schematic of the electric circuit used. (From [32]).

**Figure 5 biomolecules-14-00684-f005:**
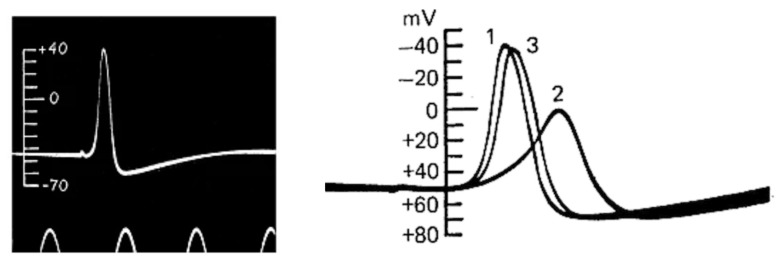
**The action potential.** (**Left**) First action potential recording from the giant squid axon by Hodgkin and Huxley in the Summer of 1939. (From [33]). (**Right**) The action potential recorded from the giant squid axon becomes slower in its rising part and the overshoot smaller when 50% of external Na^+^ is replaced with impermeant choline. Meaning of the numbers associated with the current traces 1, control; 2, 50% of external Na^+^; 3, washout. (From [34]).

**Figure 6 biomolecules-14-00684-f006:**
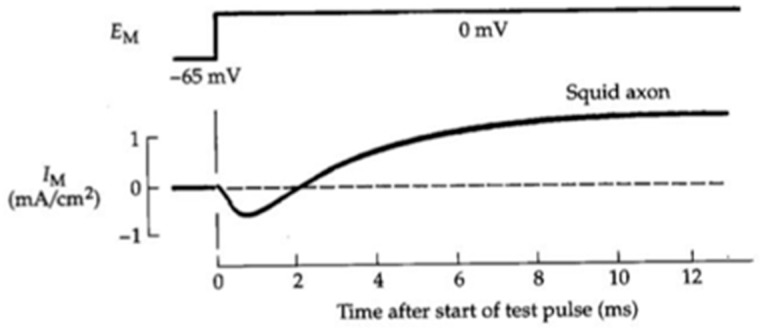
**A current recorded by Hodgkin and Huxley under voltage clamp.** The current trace shows an initial inward current followed by a delayed outward current in response to a membrane depolarization of the giant squid axon from a holding potential close to rest (−65 mV) to 0 mV. (Modified from [40]).

**Figure 7 biomolecules-14-00684-f007:**
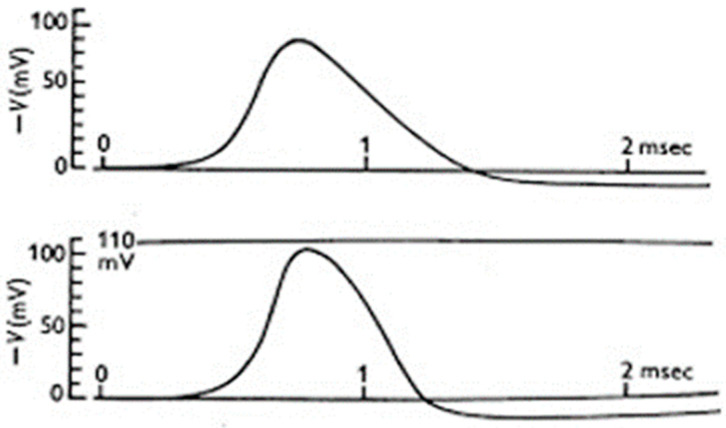
**Modelling the action potential of the giant squid axon.** Simulated (**top**) and experimental (**bottom**) action potentials. (From [38]).

**Figure 8 biomolecules-14-00684-f008:**
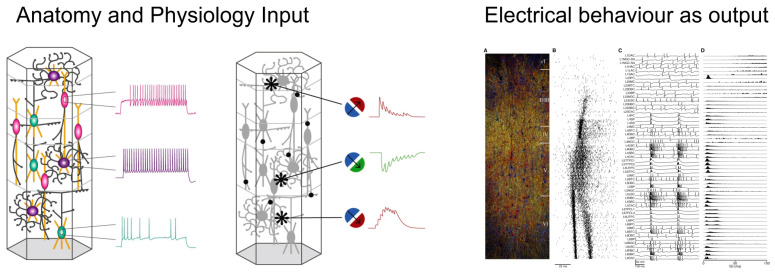
**Digital reconstruction and simulation of mouse neocortical architecture and neuronal activity.** The anatomy and physiology of neocortical microcircuitry reproduces an array of in vitro and in vivo experiments. (From [65]).

**Figure 9 biomolecules-14-00684-f009:**
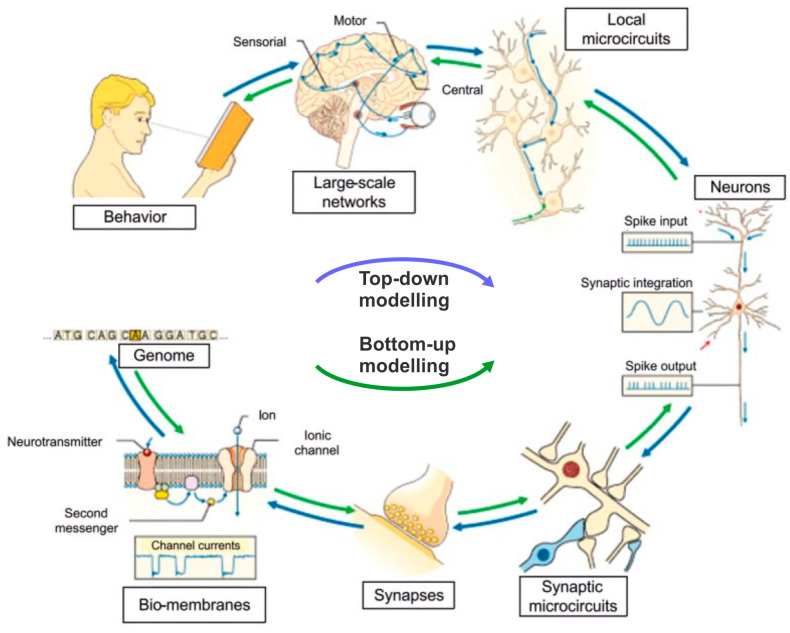
**Multiscale organization of the brain.** A full picture of the working brain requires an understanding of the numerous scales of neural organization, that is, the interplay of genes, synapses, and neurons, and how they connect into large neuronal networks and brain areas to explain the relationship among microscopic processes, large-scale brain functions, and behavior (‘bottom-up’ approach, green arrows). The reverse operation, i.e., inferring neuronal functions from ensemble measurements like those currently obtained through the use of MRI, EEG, MEG, or PET (‘top-down’ approach, blue arrows) is presently more difficult to do, yet it is a rich source of potential information. (Modified from [74]).

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
