# Peer review of "The Long Journey from Animal Electricity to the Discovery of Ion Channels and the Modelling of the Human Brain"

_biomolecules, 2024, doi:10.3390/biom14060684_

Round 1

Reviewer 1 Report

Comments and Suggestions for Authors

The present review by Catacuzzeno and colleagues details the history of research on electrical signals in living organisms. It begins with the discovery of “animal electricity” at the end of the 18th century and ends with the present efforts of modeling the human brain and constructing a digital brain.

The manuscript was a pleasure to read. It is well-written, nicely structured and gives a comprehensive description of all the prominent researchers and their efforts that paved the way to understanding electrical signals in living organisms. I have only some minor suggestions to further improve this excellent work:

·         Section 1 & 5 would considerably benefit from actual references to the sources of the authors.

·         Section 1.3 & 2.1: Refer to exactly where the information can be found instead of stating “see later” or “see below”.

·         Section 1.3: “Wissenschaft” in German is spelled with a capital W.

·         It is often difficult to follow in which years the described experiments were done or new findings were published (especially in section 2). I recommend mentioning more actual dates to make the topic easier to follow for the reader in a chronological context.

·         Figure 1.1 legend: Replace “Layden” with ” Leyden”.

·         Section 3.2: Replace “Hodgking” with “Hodgkin”.

·         Figure 5.1 is never referenced in the main text.

·         Figure 5.2 legend: Replace “Buttom-up” with “Bottom-up”.

Comments on the Quality of English Language

Some sentences suffer from grammatical issues, e.g. “The introduction, in the summer of 1936, of the giant squid axon, thanks to the suggestion of the young British zoologist John Young, who was studying cephalopods at the Zoological Station of Naples where he had noticed these large cylindrical structures of about 1 mm in diameter, which he concluded were axons.” – Further screening of the text for these errors is advisable.

Author Response

The present review by Catacuzzeno and colleagues details the history of research on electrical signals in living organisms. It begins with the discovery of “animal electricity” at the end of the 18th century and ends with the present efforts of modeling the human brain and constructing a digital brain.

The manuscript was a pleasure to read. It is well-written, nicely structured and gives a comprehensive description of all the prominent researchers and their efforts that paved the way to understanding electrical signals in living organisms. I have only some minor suggestions to further improve this excellent work:

  • Section 1 & 5 would considerably benefit from actual references to the sources of the authors.

R: Most appropriate comment, thank you. In all, we added 12 references.

  • Section 1.3 & 2.1: Refer to exactly where the information can be found instead of stating “see later” or “see below”.

R: Done

  • Section 1.3: “Wissenschaft” in German is spelled with a capital W.

R: Done

  • It is often difficult to follow in which years the described experiments were done or new findings were published (especially in section 2). I recommend mentioning more actual dates to make the topic easier to follow for the reader in a chronological context.

R: We have indicated the actual dates virtually everywhere (i.e., in eight places)

  • Figure 1.1 legend: Replace “Layden” with ” Leyden”. R: Done
  • Section 3.2: Replace “Hodgking” with “Hodgkin”. R: Done
  • Figure 5.1 is never referenced in the main text. R: Done
  • Figure 5.2 legend: Replace “Buttom-up” with “Bottom-up”. R: Done

Comments on the Quality of English Language

Some sentences suffer from grammatical issues, e.g. “The introduction, in the summer of 1936, of the giant squid axon, thanks to the suggestion of the young British zoologist John Young, who was studying cephalopods at the Zoological Station of Naples where he had noticed these large cylindrical structures of about 1 mm in diameter, which he concluded were axons.” – Further screening of the text for these errors is advisable.

We replaced the original sentence with the following one and also had a proficient English-speaking colleague read the Ms. “The solution came with the introduction, in the summer of 1936, of the giant squid axon, thanks to the suggestion of the young British zoologist John Young, who was studying cephalopods at the Zoological Station in Naples. These axons are cylindrical structures up to 1 mm in diameter into which it was easy to insert hand-made glass electrodes.”  

Reviewer 2 Report

Comments and Suggestions for Authors

In the manuscript entitled “The long journey from animal electricity to the discovery of ion channels” the authors provide a historical perspective from the discovery of animal electricity in 18th century to the ion channels in 20th century. The then authors review the main efforts to construct a digital brain in 21st century.

The review is well-structured and easy to follow. Its topic is interesting and attractive but I have some concerns, as follows:

1.    In this manuscript, the main emphasis of the second part of review is on the electrophysiology of the brain and attempts to create a supercomputer (digital brain). This information is not present in the current version of the title. Thus, it is recommended to make it more focused emphasizing the Modelling the human brain.

2.    Do the authors have permission to reprint the figures in the manuscript?

-       The colored Figure 1.1 is marked by Fine Art America. Is it correct?

3.    This historical review is partially similar to the reviews by Verkhratsky et al., 2006 PMID: 17072639 and 2014 PMID: 25023299. The Reference on Verkhratsky et al., 2006 is present only in the Figure legend 2.1 but not in the Reference list. Please add needed References.

4.    There should be more detailed information about the time when the events described occurred, and about all scientist:

- The authors write only second names of some scientist do not provide any more information about them. For instance, du Bois-Reymond (line 116), Myer and Overton (line 145), Nernst (line 152), von Helmotz (line 198), Hooke and Leeuwenhoek (line 215), Gorter and Grendel (line 223), Davson and Danielli, Hodgkin and Huxley (line 289) and further. Please indicate first names, as well as the country and institutions where they made their discoveries.

- It would be better to indicate the year or at least decade when the events occurred. Throughout the text there are phrases “some fifty years later” (line 101), “This current will be named ‘injury currents’ decades later” (line 106), “these issues arrived at the turn of the century” (line 126), “at the turn of the century” (line 145), “A few years after” (line 182), “at the turn of the century, was much better defined 25 years later” (lines 222-223), “ten years later” (line 233) etc.

5. There are no needed References in some paragraphs: Lines 69-71, 116-119, 184-186, 525-534, 679-681. Please add references.

Why the authors suggested that ideas of Galvani were influenced by the classic views of Galeno and Rene Descartes? Are there any References in literature?

Minor comments

-       On Fig 1.1 From (Galvani, 1792)- line 49 in the text 1791 – line 54. What date is correct?

-       There are numbers 1, 2, 3 on the Figure 3.2 on the right. Please indicate them in Figure legend.

-       Line 284 Hodgking

-       Please replace Na to Na+  and K to K+ (lines 322, 324, 325, 364, 373, 385, 389, 396 etc)

-       Line 427 IC need Ic

-       Line 653 that (delete repeat)

Author Response

In the manuscript entitled “The long journey from animal electricity to the discovery of ion channels” the authors provide a historical perspective from the discovery of animal electricity in 18th century to the ion channels in 20th century. The authors then review the main efforts to construct a digital brain in 21st century.

The review is well-structured and easy to follow. Its topic is interesting and attractive but I have some concerns, as follows:

  1. In this manuscript, the main emphasis of the second part of the review is on the electrophysiology of the brain and attempts to create a supercomputer (digital brain). This information is not present in the current version of the title. Thus, it is recommended to make it more focused emphasizing the Modelling the human brain.

R: We have emphasized this theme by adding those words to the Ms title.

  1. Do the authors have permission to reprint the figures in the manuscript? The colored Figure 1.1 is marked by Fine Art America. Is it correct?

R: We have already set to work to request permission to reproduce the figures. Second, we replaced Figure 1.1, marked Fine Art America, with the same b/w figure, which originally appeared in De motu muscularis of 1791.

  1. This historical review is partially similar to the reviews by Verkhratsky et al., 2006 PMID: 17072639 and 2014 PMID: 25023299. The Reference on Verkhratsky et al., 2006 is present only in the Figure legend 2.1 but not in the Reference list. Please add needed References.

R: Reference added.

  1. There should be more detailed information about the time when the events described occurred, and about all scientist:

 The authors write only second names of some scientist do not provide any more information about them. For instance, du Bois-Reymond (line 116), Myer and Overton (line 145), Nernst (line 152), von Helmotz (line 198), Hooke and Leeuwenhoek (line 215), Gorter and Grendel (line 223), Davson and Danielli, Hodgkin and Huxley (line 289) and further. Please indicate first names, as well as the country and institutions where they made their discoveries.

R: First names added everywhere.

- It would be better to indicate the year or at least decade when the events occurred. Throughout the text there are phrases “some fifty years later” (line 101), “This current will be named ‘injury currents’ decades later” (line 106), “these issues arrived at the turn of the century” (line 126), “at the turn of the century” (line 145), “A few years after” (line 182), “at the turn of the century, was much better defined 25 years later” (lines 222-223), “ten years later” (line 233) etc.

R: We have indicated the year in all the places mentioned.

  1. There are no needed References in some paragraphs: Lines 69-71, 116-119, 184-186, 525-534, 679-681[1]. Please add references.

R: Appropriate suggestion, thank you. Done

  1. Why the authors suggested that ideas of Galvani were influenced by the classic views of Galeno and Rene Descartes? Are there any References in literature?

R: It is not easy to find specific references to this. However, it is fairly accepted that the views of the ancient Greeks (e.g. Aristotle) up to the time of Galen, such as the concept of pneuma, continued to be relevant in most physiological, cosmological and even ethical questions for many centuries to come. These views were still present for Descartes in the 17th century, who reworked them as ‘animal spirits’ to explain the functions of the body, and continued through most of the 18th century, when focus shifted to items such as Galvani's nerve conduction, supposedly determined by the animal spirit. For much of the 18th century, thinkers continued to refer to the ‘animal spirits’ to explain how nerve conduction work, likely finding it preferable to stick to a possibly wrong explanation than admit ignorance for how excitation moved down the nerves. However, we have included as reference a fine paper on the subject, written by an eminent scholar who, in our opinion, excellently makes the point (Cobb, M. 2002. Exorcizing the animal spirits: Jan Swammerdam on nerve function. Nat Rev Neurosci. 3:395-400).

Minor comments

-       On Fig 1.1 From (Galvani, 1792)- line 49 in the text 1791 – line 54. What date is correct?

R: The original paper appeared as an ‘opusculum’ in the transactions of the Bologna Academy of Science, in 1791. The discrepancy has been fixed, thank you.

-       There are numbers 1, 2, 3 on the Figure 3.2 on the right. Please indicate them in Figure legend. R: Done

-       Line 284 Hodgking. R: Done

-       Please replace Na to Na+  and K to K+ (lines 322, 324, 325, 364, 373, 385, 389, 396 etc). R: Done

-       Line 427 IC need Ic. R: Done

-       Line 653 that (delete repeat). R: Done

Round 2

Reviewer 2 Report

Comments and Suggestions for Authors

Please fix typo in legend to Fig. 3.2 wontrol

Otherwise, I am satisfied with the corrections introduced by the authors.